# Pancreatic Cancer Cachexia: The Role of Nutritional Interventions

**DOI:** 10.3390/healthcare7030089

**Published:** 2019-07-09

**Authors:** Toni Mitchell, Lewis Clarke, Alexandra Goldberg, Karen S. Bishop

**Affiliations:** 1School of Medical Sciences, Faculty of Medical and Health Sciences, University of Auckland, Auckland 1023, New Zealand; 2Department of Nutrition, School of Medical Sciences, Faculty of Medical and Health Sciences, University of Auckland, Auckland 1023, New Zealand; 3Auckland Cancer Society Research Centre, School of Medical Sciences, Faculty of Medical and Health Sciences, University of Auckland, Auckland 1023, New Zealand

**Keywords:** cachexia, pancreatic cancer, nutrition, mechanisms

## Abstract

Pancreatic cancer is a cancer with one of the highest mortality rates and many pancreatic cancer patients present with cachexia at diagnosis. The definition of cancer cachexia is not consistently applied in the clinic or across studies. In general, it is “defined as a multifactorial syndrome characterised by an ongoing loss of skeletal muscle mass with or without loss of fat mass that cannot be fully reversed by conventional nutritional support and leads to progressive functional impairment.” Many regard cancer cachexia as being resistant to dietary interventions. Cachexia is associated with a negative impact on survival and quality of life. In this article, we outline some of the mechanisms of pancreatic cancer cachexia and discuss nutritional interventions to support the management of pancreatic cancer cachexia. Cachexia is driven by a combination of reduced appetite leading to reduced calorie intake, increased metabolism, and systemic inflammation driven by a combination of host cytokines and tumour derived factors. The ketogenic diet showed promising results, but these are yet to be confirmed in human clinical trials over the long-term. L-carnitine supplementation showed improved quality of life and an increase in lean body mass. As a first step towards preventing and managing pancreatic cancer cachexia, nutritional support should be provided through counselling and the provision of oral nutritional supplements to prevent and minimise loss of lean body mass.

## 1. Introduction

Cachexia is a complex, multifactorial syndrome defined by involuntary weight loss due to an ongoing loss of skeletal muscle mass and lipoatrophy [1]. This weight loss is due to metabolic changes and decreased calorie intake [1] and has a negative impact on survival and quality of life (QoL) [2]. Cachexia is commonly associated with pancreatic cancer, manifesting in approximately 70–80% of patients and contributing 33% to the pancreatic cancer mortality rate [3].

Pancreatic cancer, although not one of the most common cancers globally, is a cancer with one of the highest mortality rates [4,5]. Various treatment options are available, including surgery, radiotherapy, chemotherapy, and immunotherapy. Surgery is only suitable in about 15% of cases, i.e., where the cancer is confined to the pancreas, and therefore it is not surprising that surgery (alone or in combination) is associated with a higher success rate than other treatment options that are targeted at metastatic pancreatic cancers. Pancreatic cancer presents from the age of approximately 45 years, with two-thirds of patients at least 65 years old [4], and a 10-year survival rate of less than 1% in the UK [6].

One of the problems associated with cancer cachexia is the difficulty experienced with standardising diagnostic criteria, and therefore, cachexia in the cancer patient is often left untreated [7]. Diagnostic criteria for cancer cachexia, in general, are summarised in Table 1. However, weight loss alone is not regarded as an independent prognostic factor, and many patients present with cachexia at the time of pancreatic cancer diagnosis [1].

In 2011, an international consensus agreed to a definition of cancer cachexia as follows: “Cancer cachexia is defined as a multifactorial syndrome characterised by an ongoing loss of skeletal muscle mass (with or without loss of fat mass) that cannot be fully reversed by conventional nutritional support and leads to progressive functional impairment. The pathophysiology is characterised by a negative protein and energy balance driven by a variable combination of reduced food intake and abnormal metabolism” [8]. However, this definition has not necessarily been adopted in practice and applied to all relevant studies post-2011. Many investigators believe that cancer cachexia is resistant to nutritional interventions. In this narrative, we will present evidence that indicates that nutrition can positively influence the pancreatic cancer cachexic patient.

In addition to difficulties associated with diagnosing cachexia in pancreatic cancer patients, the mechanisms and biomarkers associated with cachexia are yet to be definitively elucidated. It is thought that weight loss associated with cachexia is seen after metabolic changes have occurred and that tumour associated proinflammatory cytokines drive these metabolic changes [10]. These mechanisms pose challenges for the reversal and management of cachexia through nutrition alone, and a systems approach might be more appropriate [8]. 

Several mechanisms that contribute to the characteristic negative energy balance seen in pancreatic cancer cachexia.This pathological state is driven by the combination of both reduced appetite and increased metabolism [1,8]. Current understanding implicates systemic inflammation, driven by both host cytokines and tumour derived factors as key underlying mechanisms of cancer cachexia pathology.

Raised serum C-reactive protein (CRP), a biomarker for systemic inflammation, interleukin 6 (IL6) [9], and growth differentiation factor-15 (GDF-15) [11] are thought to be associated with cachexia. Elevated CRP, IL6, IL8, and IL10, particularly in combination, are associated with an increase in pancreatic cancer morbidity and mortality. These cytokines may be produced by the immune system or directly from the tumour [12]. GDF-15 appears to be elevated in plasma of pre-cachexic and in cachexic cancer patients experiencing weight loss [13,14,15,16]. However, GDF-15 has not been extensively researched in association with pancreatic cancer but may be influenced by lysine-deficiency and chronic high-fat consumption [11].

The influence of cytokines in pancreatic cancer cachexia can be broadly categorised by the centrally-mediated pathways, involving the hypothalamus, or by the direct lipolytic and proteolytic effects of cytokines, collectively known as the peripheral pathways. The hypothalamus is vital for controlling the homeostatic energy balance by integrating a wide range of information regarding adiposity and energy expenditure. Increased systemic inflammation is thought to interfere with these homeostatic feedback loops by promoting the anorexigenic (satiety) pathways while simultaneously suppressing the orexigenic (hunger) pathways, thus inducing appetite suppression [17]. Cytokines not only activate the anorexigenic circuits to suppress appetite, but also increase catabolism by increasing energy expenditure. IL6 is a key cytokine which operates synergistically with tumour necrosis factor α (TNFα) to activate other cytokines to sustain a proinflammatory response. Clinical trials have shown an association between circulating IL6 and degree of both weight loss and survival rates in pancreatic cancers [12].

Metabolic dysfunction is also heavily influenced by tumour-derived cachexic factors. The best-understood factors are the lipid mobilising factor (LMF) and proteolysis-inducing factor (PIF) [18]. In in vivo mouse models, LMF was implicated in increasing lipolysis and white adipose tissue (WAT) turnover rates in order to fuel increased brown adipose tissue (BAT) cell thermogenesis, resulting in lipid depletion and increased energy expenditure [19]. Similar to the LMF, PIF was discovered in a mouse model of cachexia [20] and then isolated from patients urine. Administering PIF to mice has been shown to induce cachexia symptoms such as a loss of lean body mass without reducing food intake [21]. The mechanism behind PIF lean mass loss is thought to be two-fold, with increased protein catabolism and decreased protein synthesis [22].

In addition to systemic inflammation and tumour-derived factors, invasion by a pancreatic tumour can physically block both the pancreatic duct and parts of the gastrointestinal tract (GI). This blockage results in anorexic maladies such as pain, nausea, and impaired GI functionality, which can lead to decreased energy intake and weight loss [23].

Due to the complex synergistic interactions between mechanisms, the overall state of pancreatic cancer cachexia is greater than simply the combination of each mechanism in isolation. In this narrative, we will present evidence regarding how nutrition may interact with these mechanisms and modulate outcome and/or quality of life. There is an urgent need to establish appropriate treatment for patients with cachexia, and it is therefore important to carry out well designed, multicentre studies to assess the value of various nutritional therapies to help improve treatment outcomes in this vulnerable group of patients.

## 2. Nutritional Interventions to Support Pancreatic Cancer Cachexia Management

### 2.1. Calorie Intake

Calorie intake has been studied for quite some time in the hope of improving treatment outcomes in cancer patients. Negative energy balance contributes to weight loss more severely in cancer patients compared to healthy individuals [16]. Insufficient food consumption has been shown to contribute significantly to weight loss in cancer cachexic patients [16,24]. However, evidence supports the notion that an increased metabolic rate may also be a contributing factor. In support of this hypothesis, Warnold et al. (1978) reported that the energy expenditure of (predominantly hospitalised) cancer patients was 2020 kcal/24 h, whereas, energy expenditure averaged 1420 kcal/24 h in non-cancer patients [25]. This difference in energy expenditure could explain much of the chronic weight loss experienced by cancer cachexic patients. Measurements of calorimetry, energy balance, body composition, and energy requirements for nitrogen equilibrium have identified abnormally high energy expenditure in 50–60% of hospitalised cancer patients [26]. In a recent appraisal in Nature Reviews, Baracos et al. (2018) also identified increased metabolism and decreased energy intake as contributing to weight loss in cachexic patients [27]. 

Caloric requirements of patients with cancer need to be established based on the patient’s nutritional demands. This includes increased energy expenditure due to the energy requirements of the tumour, as well as the energy demands that rise due to the body’s reaction to the presence of the tumour. Despite increased energy needs, reduced caloric intake is common in patients with cancer cachexia. Sensory abnormalities (altered taste or smell) have been considered as a contributing factor, but results from studies designed to address this, have lacked statistical significance [26]. Even though nutritional support may not cure cancer, it can ‘cure’ malnutrition and may be beneficial in improving the patient’s QoL. Nutritional therapy is used as repletion or maintenance. Decreased morbidity and mortality has been seen in patients consuming 35 to 40 kcal/kg/day of non-protein energy and 0.25 to 0.30 g/kg/day of nitrogen [28]. There is a tendency towards shorter survival in patients with a lower energy intake [29], and higher calorie intake equates to longer survival [30]. Therefore, there is a relationship between calorie intake and survival.

Fats provide more calories per gram than proteins or carbohydrates and fat is an excellent method of delivering calories to a malnourished person. Fat is an essential fuel to provide energy for the heart and brain, at a minimal caloric cost. In contrast, calories stored in muscles are associated with increased energy utilisation and therefore increased caloric need. However, fat is a potent releaser of cholecystokinin that is present in higher levels following a meal and produces satiation [31]. These nutritional guidelines should not only improve nutritional status but also prevent further deterioration. Enhancement of the activity of adjuvant therapies remains the main goal. 

### 2.2. Oral Nutritional Supplementation and Parenteral Nutrition

A cachexic patient is likely to consume a diet that would be insufficient to maintain weight in a healthy individual [32], although it is acknowledged that in many patients it is the increase in metabolism that creates the need to increase calorie intake [25]. It is therefore critical to raise the energy content of a diet consumed by a cachexic patient. Oral nutritional support should focus on increasing energy intake via a change in diet, and oral nutritional supplementation. Upon diagnosis of pancreatic cancer cachexia, the patient should be given nutrition counselling about ways that they can increase their energy and protein intake as higher energy intake has been associated with improved QoL and survival [29,30,33]. Increasing the size of meals, the number of snacks, or the energy value of existing meals, using energy-dense foods and protein are good ways to increase overall energy intake, depending on what the patient can tolerate. Studies investigating the impact of nutritional counselling, to increase energy intake at diagnosis, on cachexia, are lacking. However, it would seem to be a prudent first step to attempt to delay or slow cachexia development, hence improving the QoL of patients.

If a patient continues to lose weight after dietary changes, then an oral nutritional supplement (ONS) may be of value. Solheim et al. [34] suggest that nutritional counselling in combination with non-steroidal anti-inflammatory drugs, exercise support, and ONS allowed weight stabilisation compared to the controls, even with only 48% compliance of the ONS. Bauer et al. [35] performed a post hoc analysis (Table 2) wherein they tentatively suggested that if patients were compliant with consuming an ONS, then there was a tendency for weight improvement and improved QoL. While the studies were not adequately powered to show significant results or associations in post hoc analyses, they show evidence that merits further investigation. 

In patients who are unable to swallow or experience difficulties eating (due to anorexia or physical obstructions, amongst other reasons), parenteral nutrition (PN), which is characterised by the delivery of nutrient-rich solutions into the vein, may be the only option. PN raises challenges as it may cause allergic reaction at the site of IV-line administration, does not stimulate gut motility due to direct administration of the nutrients into the blood circulatory sytstem, carries the risk of infection and has been shown to induce hyperglycaemia [36]. Appropriate use of parenteral nutrition in oncology was fiercely contested due to a lack of robust large scale trials [37]. However, since this time, a number of clinical trials have been published showing benefit to the majority of patients with pancreatic or advanced cancers receiving PN [38,39,40].

### 2.3. Ketogenic Diet

The ketogenic diet (KD) is a dietary pattern that has been proposed as a therapeutic option to prevent tumour progression or alleviate symptoms of cachexia [49,50]. In contrast, concern has been expressed that it will trigger or exacerbate cachexia development [51]. 

A KD is a high fat, low carbohydrate diet which leads to elevated levels of ketone bodies in the blood. Most tumours are unable to metabolise ketone bodies and rely on glucose as an energy source. In contrast, healthy cells can use ketone bodies for fuel, so it is thought that the KD can limit the energy source of the tumour, while still providing fuel for the host [49,50,52,53]. Preclinical and animal studies have suggested that ketone bodies may inhibit pancreatic cancer cell growth, thus reducing tumour size and cachexia through inhibiting loss of muscle and body weight [50]. 

In vitro Shukla et al. [50] showed that Capan1 and S2-013 pancreatic cancer cell survival was inhibited by ketone bodies at 10 and 20 mM after 72 h of treatment, with results also suggesting that ketone bodies may inhibit cancer-induced cachexia. 

Pancreatic cancer mice models have shown significantly lower tumour weight and volume and glucose concentration (*p* < 0.05), while β-hydroxybutyrate (βHB) (ketone body) level, muscle weight, and carcass weight were significantly higher (*p* < 0.05) in the mice fed a KD versus standard diet (SD) [50]. In this study, a 45% increase in muscle weight and 20% increase in carcass weight was observed in mice fed the KD compared to a SD [50]. The authors suggest that the KD may be associated with diminishing tumour growth and inhibition of cancer-induced cachexia through metabolic changes [50]. The KD has also been shown to prolong the mean survival time in mice fed a KD [54], compared to a SD. A lower tumour burden, with slower tumour growth [54,55,56] and size [55], with less metastatic spread to the brain, lungs, liver, kidneys, spleen, and adipose tissue [55], were also observed in mice fed a KD.

Evaluation of a KD in cancer patients has shown that the diet appears to be well tolerated [53,57,58,59], with a few diet-related side effects [53,57,58,59] including constipation [53,57,58], fatigue [53,57,58], and leg cramps [57] which were all reversible. Hypoglycaemia was not observed [60]. Weight loss was low [53,57,58] unless a restricted KD was implemented in overweight patients or weight loss was intended [53,58]. Changes in body composition showed decreases in fat stores drove the weight loss, while muscle mass was preserved [59,61]. Small improvements in QoL have also been observed [53,61]. Recent research has shown that a high-fat diet, and lysine-deficiency in mice can increase GDF-15 levels [11], and lead to appetite suppression. These preliminary results need to be confirmed in humans and suggest that it may be important for cancer patients to consume adequate lysine to help avoid or reverse cachexia. In contrast, the long-term consumption of a high-fat diet, commonly associated with a KD, may promote cachexia due to the increase in GDF-15 [11]. Further research on the role of GDF-15 and lysine/fat consumption, in the context of pancreatic cancer cachexia, is required.

Conclusions regarding progression-free survival are difficult to determine due to varying cancer diagnoses and time periods, small patient numbers, and in some cases, an advanced disease leading to death. There does, however, appear to be a beneficial relationship between ketosis and cancer, Fine et al. showed a 3-fold higher level of ketosis correlated to stable disease or partial remission [57]. The results on the KD are not conclusive, and the applicability of this diet in pancreatic cancer cachexic patients is yet to be fully understood. A diet that is well tolerated with few side effects that appears to preserve muscle mass could prove to be instrumental in limiting cachexia development, but further work is required to elucidate this link. Investigations at an earlier stage in disease progression are warranted to evaluate long term palatability and effect of the KD in mitigating pancreatic cancer cachexia. 

### 2.4. Fish Oil 

Fish oil (FO) is widely known for its anti-inflammatory properties, yet patients with advanced cancer have been shown to have low concentrations of (n-6 or n-3) polyunsaturated fatty acids (PUFAs) in their phospholipids of plasma (<30% of normal values) [42,43]. Increasing interest in the properties of essential n-3-PUFAs or omega-3 fatty acids (ω-3FA) from oily fish has resulted in a vast increase in publications. Their properties (Figure 1) are involved in the synthesis of cell membranes, receptors, eicosanoids (prostaglandins and leukotrienes), and functioning enzymes. Constituent anti-inflammatory ω-3FA, namely eicosapentaenoic acid (EPA) and docosahexaenoic acid (DHA), have been associated with a reduction in platelet aggregation, reduced blood viscosity, and production of an anti-inflammatory response [62]. 

EPA, in particular, has been shown to block ubiquitin-proteasome induced muscle proteolysis, which could be suggestive of its favourable effects on muscle conservation in wasting syndromes. Additionally, populations whose diets have high amounts of ω-3FA demonstrate the lowest occurrences of cancer and animal studies have shown that ω-3FAs has the potential to reduce tumour growth [63,64,65,66]. 

Administration of FOs showed a reduction of cytokine production in animals and healthy subjects and an increase in cultured cancer cell death [63]. The ω-3FAs compete with Arachidonic acid (AA) for the biosynthesis in the active site of lipid-second messengers [67]. This displacement of AA (by EPA or DHA) in the phospholipid component of cellular membranes results in a reduced inflammatory response induced by pro-inflammatory cytokines and may, therefore, reduce cancer cachexia (34). Several mediators of cachexia are affected by FO and EPA. Recent studies have confirmed the effects of ω-3FA on weight stabilisation and on the reduction of pro-inflammatory cytokines [43,68,69,70]. Colomer et al. [68], performed a meta-analysis and concluded that ω-3FA stabilised the effect of weight, appetite, and QoL in advanced cancer patients, with a minimum dose of 1.5 g per day over at least eight weeks. 

ω-3FA and their effect on pancreatic cancer cachexia (Table 2) have shown a trend towards stabilisation or an increase in weight [35,43,47] as well as an increase in lean body mass [35,42]. However, there were no significant differences detected in CRP, a measure of inflammation [42,47,48]. A significant decrease in IL6 production and a fall in the proportion of patients excreting PIF were seen after ONS containing ω-3 was consumed, suggesting that ω-3 may normalise some of the metabolic changes that prevent weight gain [44].

In two studies, a 45% drop-out rate due to non-compliance, progressive disease, gastrointestinal side effects, or death was experienced [42,43]. Such a high drop-out rate weakens the validity of the results but is unavoidable and a common problem with pancreatic cancer studies of this nature. A high proportion of studies delivered the ω-3 dose as part of an ONS without a control arm. The protein intake from the ONS varied due to compliance. In these studies, it is suggested that ω-3 could aid in weight stabilisation or gain, but further work is required to elucidate the benefits of ω-3 intake independent of increased protein consumption. In two separate studies investigating ω-3, one via capsule dosing, the other via an ONS, both with an appropriate control, failed to demonstrate a weight stabilising effect of ω-3FA [45,46]. In general, compliance was poor with either a lower consumption than the planned dosage, in some cases only 50%, or dropout (up to 45%), which may have affected the result.

Patients consuming FOs often develop GI side effects such as ‘fishy’ regurgitation and taste, nausea, or flatulence if the dose exceeds the minimal effective dose of 1.5 g n-3 FA daily [71,72]. To resolve poor compliance due to GI side effects, the use of marine phospholipids (MPL) has been investigated. MPL rapidly disperse in the GI-tract [73], while in comparison FOs form larger droplets and settle on the top of the gastric juice; thus, MPLs are digested more quickly, and this may explain the reduced GI side effects. 

Intervention studies carried out by Ramprasath et al. [74], and Ulven et al. [75] showed better bioavailability of MPLs, due to higher integration of EPA and DHA into the cellular membranes (when given as phospholipids instead of triglycerides). Taylor et al. [76], conducted a study on 17 cachexic cancer patients over six weeks by using MPL extracted from salmon roe, which contains 30% EPA and DHA. The compliance in this study was high at 94%  ±  2% of prescribed MPL capsules taken three times per day (500 mg soft gel capsule). The weight of these patients was shown to stabilise after supplementation of low doses of MPL (0.3 g ω-3 FA/day), with improvement in appetite and QoL. Levels of CRP decreased significantly during the intervention. Werner et al. [47] compared FO and MPL consumption in a randomised trial where pancreatic cancer patients consumed 0.3 g ω-3 FA daily. Both FO and MPL resulted in weight stabilisation compared to pre-study weight. MPL capsules had fewer side effects and were better tolerated than the FO capsules [47,76].

FO and MPL appear to show a positive trend towards weight stabilisation or gain, and appetite improvement in individuals with pancreatic cancer cachexia [42,43,47,76].

### 2.5. Amino Acids

β-hydroxy-β-methylbutyrate (HMB), a metabolite of leucine, as well as glutamine and arginine, have been implicated in attenuation of muscle loss associated with cancer cachexia. In murine myotubes, PIF induced protein breakdown was inhibited by HMB at a concentration of 50 µmol/L [78]. HMB was also found to attenuate lipopolysaccharides [79], TNF-α and interferon-γ (IFN-γ) [79,80], and angiotensin II inhibited protein synthesis [79]. The attenuation of inhibition of protein synthesis appears to be via multiple mechanisms. Mechanisms proposed include stimulation of protein synthesis by HMB through the mTOR/p70S6 k pathway [78], attenuation of the ubiquitin-proteasome proteolytic pathway [80,81], and attenuation of the protein degradation pathway which involves activation of caspases-3 and -8, autophosphorylation of RNA dependent kinase and formation of reactive oxygen species [80].

In murine models’ supplementation with HMB caused a significant decrease in weight loss compared to tumour bearing (TB) controls [82,83], and a reduction in tumour weight [83,84]. Nunes et al. [84] showed a 40% decrease in the tumour weight (*p* < 0.05) in rats and weight gain of 19.5 ± 2.4 g in the TB group supplemented with HMB (compared to weight loss −3.3 ± 1.6 g TB control) (*p* < 0.05). Supplementation with HMB resulted in a 29% increase in rat carcass weight in non-TB control groups [84]. Supplementation with HMB also showed a trend towards longer survival time, with subcutaneous TB group doubling survival time to 28 days (*p* < 0.05), while the intraperitoneal TB group showed a 42% increase (NS), possibly indicating that HMB needs time to effect a change [85].

Promising results have been seen in clinical studies using a combined HMB/glutamine/arginine formulation compared to controls. May et al. [86] showed significantly increased body weight, and fat-free mass (FFM) after 24 weeks supplementation with an HMB/glutamine/arginine supplement, compared to an isonitrogenous, isocaloric supplement of alanine, glutamic acid, glycine, and serine. Body weight increased 2.27 ± 1.17 kg (*p* = 0.06 vs. baseline) in the treatment group, while the controls showed a non-significant small increase (0.27 ± 1.39 kg) [86]. FFM also showed a significant increase in the treatment group, 1.6 ± 0.94 kg compared to the controls 0.48 ± 1.08 kg (*p* = 0.02) (37). Rathmacher et al. [87] similar study compared supplementation with HMB/glutamine arginine in healthy volunteers (*n* = 38), patients with AIDS (*n* = 68) and cancer (*n* = 49) with a matching control group. In cancer patients, an increase in net lean mass of 2.46 kg (*p* = 0.02) and red blood cell formation, including increased haemoglobin and haematocrit, was observed [87]. There was a perceived decline in muscle weakness in the intervention group, but not in the control group, which may be indirectly related to the increase in muscle mass associated with the treatment [87]. However, Berk et al., in a randomised controlled trial (*n* = 472), were unable to find a significant change in lean body mass in cachexic patients given an HMB/glutamine/arginine supplement [88]. These results should, however, be viewed with caution as dropout was high in all studies, approximately 60–80% of participants [86,87,88]. Higher dropout rates were seen in the control groups indicating that they are unlikely treatment related. Gastrointestinal side effects were a commonly cited reason for drop out, as were time constraints, effort to mix and take the supplement, poor appetite, and some patients just refused to complete treatment.

### 2.6. Branch Chain Amino Acids

Muscle catabolism associated with cancer cachexia is considered to lead to functional impairment and a decrease in quality of life [62]. It has been suggested that branch chain amino acids (BCAAs) such as leucine and valine may have the potential to counter protein catabolism seen in cachexia. Eley et al. [89] showed that valine and leucine moderated the fall in protein synthesis in response to PIF in murine myotubes, and suppressed body weight loss, increased protein synthesis and inhibited tumour growth in TB mice. Leucine also increased soleus muscle net weight and decreased protein degradation in skeletal muscle [89]. In contrast, Peters et al. [90] found that leucine supplementation in TB mice had not affected body weight loss or tumour weight compared to TB controls. However, skeletal muscle mass was higher in the TB group consuming a high leucine supplement (8 g leucine per kilogram of feed) compared to the TB group fed control chow [90]. In overweight mice, Liu et al. [91] found that leucine supplementation enhanced pancreatic tumour growth due to inhibition of glucose clearance, allowing more circulating glucose available to the tumour.

Nine malnourished cancer patients were given a BCAA total parenteral nutrition (TPN) supplement leading to increased BCAA levels in circulating plasma [92]. The results showed improved protein utilisation compared to standard TPN [92]. There appears to be a net anabolic effect on skeletal muscle of BCAA in murine models, caution is recommended in a clinical setting due to the possibility of leucine enhancing tumour growth. Further in vivo work is required to understand the BCAA anabolic effect on skeletal muscle.

### 2.7. L-Carnitine

In cancer patients, fatigue is pervasive and has a considerable impact on their QoL [93]. Decreased calorie intake, along with medication and treatment, can interfere with absorption, synthesis and excretion of L-carnitine [93]. L-Carnitine has received much attention to target pancreatic cachexia symptoms due to its key role in lipid metabolism, as it is required for the transport of fatty acids into mitochondria for β-oxidation [94,95]. Therefore, carnitine deficiency may predispose a patient to develop fatigue via impaired utilisation of long chain fatty acids for energy [93]. There is evidence to suggest that during cachexia, L-carnitine deficiency is proportional to the severity of symptoms, not only implicating it in disease pathology but, highlighting potential use as a biomarker candidate for cachexia [94].

Two studies to test safety and tolerability along with the establishment of a dose range found that carnitine was deficient in over 75% of cancer patients [93], but could be restored in 85% of cancer patients in one week [93]. It has been suggested that doses up to 3 g are well tolerated by advanced cancer patients with few adverse effects attributed to supplementation with L-carnitine [93,96] (Table 3).

Improvement in patients fatigue symptoms [93,97,98], sleep quality [93,97], depressive symptoms [97], mood [93], an increase in lean body mass (LBM) [98], and appetite [97], weight gain, improved body composition, and reduced malnutrition [99] have all been reported as a result of 3 to 6 g/day L-carnitine supplementation (shown in Table 3).

In contrast, one randomised and two randomised double-blind, placebo-controlled trials failed to show a significant effect on fatigue [96,100], LBM [100], or appetite [100]. Contributing to these results may have been participants with advanced illness [96], a lower dose of L-carnitine (2 g) [96,101] compared to trials showing positive results, 25% to 30% of participants failing to complete the assessments [101], and only 30% of the participants having a carnitine deficiency [101].

Patients with pancreatic cancer showed promising results when supplemented with 4 g/day of L-carnitine [99] (Table 3). Supplementation was well tolerated and led to weight gain in treated participants (BMI increase of 3.4% ± 1.35), while the placebo patients lost weight (BMI decrease of 1.5% ± 1.4 (*p* < 0.018)) [99]. Both body cell mass, and body fat, contributed to the increase in weight in the L-carnitine group [99]. L-carnitine improved cognitive function, global health status, and gastrointestinal symptoms while participants on the placebo deteriorated [99]. No significant differences were detected concerning fatigue, survival, and length of hospital stay [99].

Taken together, the findings from these studies suggest that L-carnitine could effect a change in malnutrition and body composition of the cachexic patient at higher doses. There is some concern, however, that L-carnitine may affect tumour growth, but this has not been measured [99]. Larger trials to elucidate L-carnitines effect on malnutrition and body composition, as well as the ideal dose is required. Work to alleviate concerns over L-carnitine’s impact on tumour growth is also required before clinical use can be recommended. 

### 2.8. Pancreatic Enzyme Replacement

In addition to the aforementioned nutritional interventions, pancreatic enzyme supplementation may be required to improve the absorption of nutrients. Pancreatic exocrine insufficiency (PEI) is a known consequence of diseases leading to a loss of pancreatic parenchyma, obstruction of the main pancreatic duct, decreased pancreatic stimulation, or acid-mediated inactivation of pancreatic enzymes [102]. Pancreatic cancer patients are also prone to PEI which causes symptoms such as steatorrhea, malnutrition, and weight loss, however pancreatic enzyme replacement therapy (PERT) is not often prescribed to patients with pancreatic cancer [103]. Abe et al. [48] showed that when pancreatic enzymes were administered from day 28, with an ONS fortified with ω-3, the absorption of EPA was significantly improved both from baseline and day 28 (*p* = 0.021) to day 56 (*p* = 0.011), suggesting that the administration of PERT promoted the increased absorption. Prophylactically prescribing PERT to patients with unresectable pancreatic cancer may enhance uptake of essential nutrients and help to minimise cachexia.

## 3. Limitations

Several limitations must be acknowledged. Firstly, we have presented a narrative rather than a systematic review or a critical appraisal. Study design, sample numbers, length of study, and a summary of results are presented in Table 2 and Table 3. The data presented in these tables predominantly revolves around FOs, ONS, and L-carnitine interventions due to the availability of information. Also, limitations exist around the study interventions themselves. These limitations include the high drop out rate caused by high physical and emotional burden and mortality. High physical and emotional burden also makes cancer cachexia patients less likely to participate in clinical trials [1,2,3]. The application of a standardised definition of cachexia; the accuracy of assessment of energy expenditure; as well as poor adherence and lack of accurate assessment of adherence to the intervention, also limits the strength of study findings. Furthermore, study endpoints vary, for example, one study might only consider survival, whilst another might consider QoL. Future studies should include the following:be sufficiently powered to accommodate a large dropout rate;multi-centred to ensure sufficient enrolment over a relatively short time and to ensure results are generalisable across populations;provide plenty of support for study participants to minimise drop out;include single and complex interventions;and finally, utilise several different tools to strengthen the data on dietary intake and adherence to the intervention.

## 4. Conclusions

As mentioned above, cancer cachexia patients find it difficult to comply or drop out of the intervention studies, thus making it problematic to analyse the efficacy of nutritional therapies. Also, the multifactorial nature of cachexia, as well as metabolic dysregulations associated with cancer, may mean that significant conclusions are difficult to draw.

Promising results for the KD, amino acids, and L-carnitine have been seen in vitro or animal models, but when evaluating in cancer patients’, conclusions are more difficult to define. While the KD appears to be well tolerated, it is too soon to propose it as a therapeutic option for treating cachexia. In pancreatic cancer, a KD could be utilised to prevent, slow or limit cachexia development due to the poor prognosis of the disease, but further work is required to evaluate the effect of the KD over the longer term and alleviate weight loss concerns. Contradictory results have been seen for amino acids, some trials showing increases in body weight and fat-free mass, while others were unable to find significant differences, possibly due to the significant dropout rate in these trials. Caution around leucine is also required due to the possibility of enhancement of tumour growth. L-carnitine at doses at or over 4 g/day have shown increases in lean body mass and QoL, but 2 g/day failed to show significant improvements.

FO, via its ω-3 content, has shown trends towards weight increase or stabilisation and appetite improvement in cachexic pancreatic cancer patients either via an ONS or capsule dosing. Participants were required to consume between 12 and 18 capsules (except for one study) for a maximum of 12 weeks (6 to 18 g EPA), and hence long term viability of such a dose needs to be determined. The ONS also contained protein and in randomised controlled trials using an identical supplement without the ω-3, they failed to find an advantage in ω-3 supplementation. Marine phospholipids have also shown weight stabilisation effects, and appear slightly better tolerated than FO capsules, but further work is required to elucidate the effect more clearly.

To conclude, nutritional support via counselling to increase the energy and protein content of the diet should be considered as a first step. Following this, oral nutritional support via a supplement could be considered to increase an individual’s energy and protein intake if they are unable to achieve this increase through diet alone, and PN should be implemented as soon as possible when required.


**Funding:**


## Figures and Tables

**Figure 1 healthcare-07-00089-f001:**
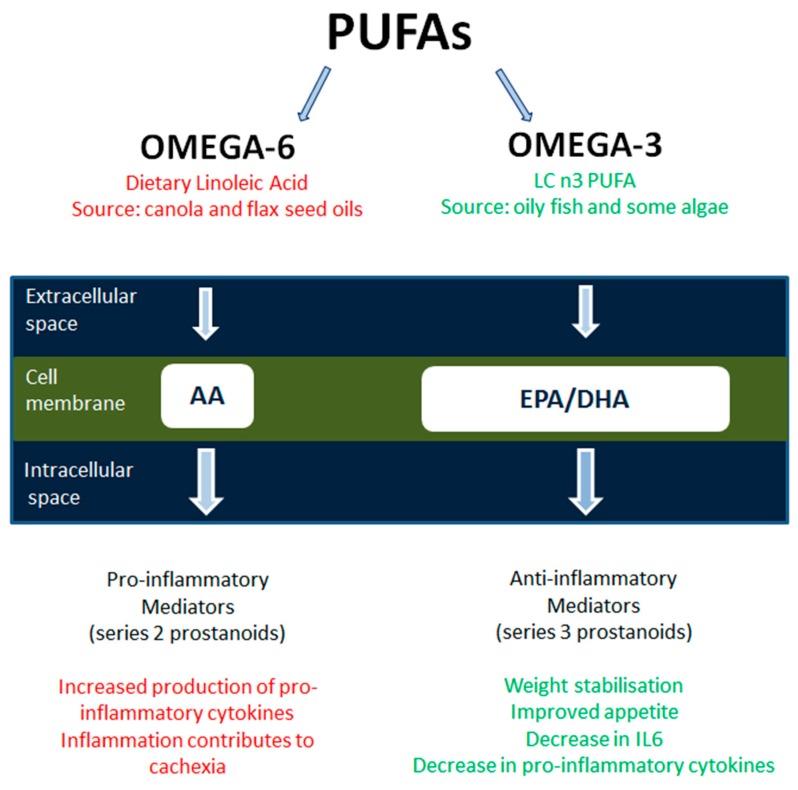
Essential n-3 and 6-PUFAs or omega-3 fatty acids (ω-3FA) from dietary linoleic acid or oily fish (respectively). Arachidonic acid derives from the cleaving of phospholipids, and it contributes to cachexia symptoms by causing inflammation [77]. It has been hypothesised that the pro-inflammatory AA ω-6FA can be partly replaced by docosahexaenoic acid and eicosapentaenoic acid and stabilise weight and reduce pro-inflammatory cytokines produced by AA [68]. AA—Arachidonic Acid; DHA—docosahexaenoic acid; EPA—eicosapentaenoic acid; IL-6—Interleukin 6; LC—long chain; PUFA—Poly Unsaturated Fatty Acid.

**Table 1 healthcare-07-00089-t001:** Criteria used to define cancer cachexia.

Criteria	Time	Citation
Involuntary weight loss >5%	6 months	[8]
BMI < 20 kg/m^2^ and any degree of involuntary weight loss > 2%	Not stated	[8]
Sarcopenia with weight loss > 2% (An appendicular skeletal muscle mass index > 7.26 kg/m^2^ and > 5.45 kg/m^2^ in males and females respectively, as defined by DEXA)	Not stated	[8]
Weight loss ≥ 5% in the presence of underlying illness, and three of the following criteria: decreased muscle strength, fatigue, anorexia, low fat-free mass index or abnormal biochemistry (CRP, IL6, anaemia, serum albumin)	≤12 months	[9] reported in [7].

Abbreviations: BMI—body mass index; CRP—C-reactive protein; DEXA—dual-energy X-ray absorptiometry; IL6—interleukin 6.

**Table 2 healthcare-07-00089-t002:** Summary of studies involving pancreatic cancer patients receiving fish oil intervention.

Refs	Study Design	Cancer Type	Nutritional Status	Comparison Details	Duration	Summary of Results	Adverse Effects
[41]	Phase II study	18 Pancreatic cancer patients	Progressive weight loss	1 g soft gel capsule; Initial dose 2 g/day, increasing to 16 g/day FO		Patients tolerated 12 g/day FO, equivalent 2 g EPA. Weight +0.3 kg (0 to 0.5) ^a^/month compared to −2.9 kg (2 to 4.6) ^a^/month pre-study (*p* < 0.002)	Offensive tasting reuctations, transient diarrhoea
[42]	Intention to treat	20 Pancreatic cancer patients	Unresectable adenocarcinoma of the pancreas	ONS (2 cans) providing 610 kcal, 32.2 g protein, 2.2 g EPA, 0.96 g DHA	7 weeks (*n* = 13)	Consumed 1.9 cans/day (1.2 to 2);Weight +2.0 kg (−0.4 to 4.6) ^a^ (*p* = 0.033); LBM + 1.9 kg (1.0 to 3.0) ^a^ (*p* = 0.0047); KPS increased 10 (0 to 10) ^a^ (*p* = 0.046)	Steatorrhoea (3)—treated with pancreatic enzyme supplementation
[43]	Prospective interventional study	23 unresectable pancreatic cancer; 3 unresectable ampullary cancers	WHO PS-2, 13% weight loss in 4 months, BMI 23.2 (21.1–27.4)% ^a^	EPA capsule 500 mg Week 1:1 g; Week 2:2 g Week 3:4 g; Weeks 4–12:6 g	12 weeks	Weight: Pre-study −2 kg (*n* = 26), week 12 +0.3 kg (*n* = 14 (*p* < 0.005 vs. week 0)	Nausea Steatorrhea
[44]		20 Pancreatic cancer patients	Unresectable adenocarcinoma of the pancreas with evidence of cachexia	ONS (2 cans) providing 610 kcal, 32.2 g protein, 2.2 g EPA, 0.96 g DHA	3 weeks	Consumed 1.9 cans/day (1.25 to 2); Weight +1.0 kg (−0.1 to 2.0) ^a^ (*p* = 0.024); Significant fall in IL6 (*p* = 0.015); Detected PIF excretion fell from 88% to 40% of patients (*p* = 0.008)	
[45]	Double blind placebo-controlled study	60 advanced cancer patients	Advanced cancer, anorexia; weight loss >5% pre-illness body weight	A daily dose of 18 gelatine capsules containing: 1000 mg fish oil (180 mg EPA, 120 mg DHA, and 1 mg vitamin E) or 1,000 mg olive oil.	2 weeks	Ability to tolerate 18 capsules was limited (9.8 ± 4 ^b^ FO, 9.2 ± 3 ^b^ PC) A strong trend towards appetite improvement in both groups	Vomiting, belching, nausea, diarrhoea, hematemesis, abdominal pain
[46]	Double-blind randomised multicentre trial	200 unresectable pancreatic cancer patients T *n* = 95; PC *n* = 105	>5% of their pre-illness weight over the previous six months, KPS ˃ 60, life expectancy > 2 months.	ONS two cans/day of 480 mL, 620 kcal, 32 g protein and 2.2 g EPA; Control-identical supplement without ω-3 FA and antioxidants	8 weeks	Intake averaged 1.4 cans/day in both groups; No net gain of LBM. Significant intervention weight loss −0.25 kg/mth (*n* = 88), control weight loss −0.37 kg/mth (*n* = 97)	
[47]	Randomised controlled double-blind trial	60 pancreatic cancer patients *N* = 31 (FO) *N* = 29 (MPL)	Weight loss ˃ 5% KPS ˃60%	Capsule either FO or MPL 1 capsule 3 times per day with meals, Equating to ω-3 300 mg/day via MPL or FO (FO capsule included 200 mg Medium chain triglycerides to balance ω-3 FA versus MPL)	6 weeks	Appetite stabilised–↑ meal portions (FO *p* = 0.02, MPL *p* = 0.05); Weight stabilisation vs. pre-study (FO *p* = 0.001, MPL = 0.003); 50% FO group and 47% of MPL group gained weight; DHA and EPA increased significantly in the blood	FO group-pyrosis, “fishy” regurgitation, loss of appetite and diarrhoea; MPL group-diarrhoea
[48]		Pancreatic (*n* = 19) and bile duct (*n* = 8) cancer patients	Patients who underwent chemotherapy Nov 2014 to Nov 2016	Enteral 2–4 packs (200 kcal and 300 mg ω-3 per pack) + pancreatic enzymes weeks 4 to 8	Baseline, 4–8 weeks	Average 2.42 ± 0.4 ^b^ packs consumed per day; ↑ trend for weight (NS), ↑ skeletal muscle mass at week 8; EPA/AA ratio improved week 8; (pancreatic enzymes aided absorption of ω-3FA).	

Abbreviations: AA—Arachidonic acid; BMI—Body mass index; DHA—docosahexaenoic acid; EPA—eicosapentaenoic acid; FO—fish oil; KPS—Karnofsky Performance Status; LBM—Lean body mass; MPL—Marine phospholipids; ω-3 FA—Omega-3 fatty acids; NS—Not significant; ONS—Oral nutritional supplement; PC—Placebo; PIF—Proteolysis-inducing factor; T–treatment; WHO PS—World Health Organization performance status. ^a^ Median and interquartile range; ^b^ Mean ± standard deviation.

**Table 3 healthcare-07-00089-t003:** Summary of L-carnitine intervention studies carried out on cancer patients.

Refs	Study Design	Characteristics	Health Status	Details	Duration	Summary of Results	Adverse Effects
[97]	Up–down dose finding design	15 cancer patients 2 dropped out	Moderate to severe fatigue Carnitine deficiency KPS ± 50	Oral Beginning dose of 250 mg/day, increased in 500 mg increments to 3000 mg	1 week	LC deficiency and self-reported fatigue 83%; LC supplementation safe up to 1750 mg/day; LC increased from 30.0 ± 6.9 ^a^ to 41.0 ± 12.1 ^a^ (*p* = 0.01); BFI improved 73 (46, 82) ^b^ vs. 50 (3, 82) ^b^ (*p* = 0.009); CES-D improved 31.3 (16, 48) ^b^ vs. 22.0 (6, 40) ^b^ (*p* = 0.028); ESS improved 17.5 (0, 24) ^b^ vs. 8 (0, 15) ^b^ (*p* = 0.015)	None reported by participants
[98]	Single arm	12 cancer patients	Locally advanced/metastatic disease with high levels of fatigue/ROS	Daily 6 g LC, in 3 doses of 2 g each Concomitant chemotherapy	4 weeks (t2)	Fatigue decreased: MFSI-SF: 12.05 ± 12.56 ^a^ (*p* < 0.001); QoL improved: Part A QoL-OS:36.80 ± 15.7 ^a^ (*p* < 0.05); EQ-5DVAS: 73.33 ± 12.4 ^a^ (*p* < 0.001); LBM increased t0:38.0 ± 7.36 ^a^ t2:40.39 ± 8.55 ^a^ kg (*p* < 0.05); Appetite increased: t0: 4.75 ± 2.59 ^a^ t2: 6.83 ± 1.9 ^a^ *p* = 0.001;	None reported that could be related to L-carnitine
[93]	Phase I/II open label clinical trial to test safety, tolerance and establish safe dose range	27 cancer patients	Moderate to severe fatigue (KPS ≥ 50) and carnitine deficiency	250 mg/day increased in 500 mg increments to 3000 mg	1 week	Total Carnitine rose from 32.8 ± 10 ^a^ to 54.3 ± 23 ^a^ (*p* < 0.001); Haemoglobin declined from 12.2 ± 2.1 ^a^ g/dL to 12.0 ± 1.8 ^a^ g/dL (*p* = 0.03); BFI improved 66.1 ± 12 ^a^ vs. 39.7 ± 26 ^a^ (*p* < 0.001); ESS improved 12.9 ± 7 ^a^ vs. 9 ± 6 ^a^ (*p* = 0.001); CES-D improved 29.2 ± 12 ^a^ vs. 19 ± 12 ^a^ (*p* < 0.001); Supplementation well tolerated upto 3000 mg;	Mild nausea (2)
[96]	Double-blind placebo-controlled, then open-label phase	29 advanced cancer patients treatment 17; placebo 12	Moderate to severe fatigue (KPS ≥ 50) and carnitine deficiency	0.5 g/day (2 days), 1 g/day (2 days), 2 g/day (10 days)	2 weeks placebo-controlled 2 weeks open label	Total Carnitine increased from 32.9 ± 3.8 ^a^ to 56.6 ± 20.5 ^a^ (*p* = 0.004); Analysis from the 14-day blinded phase failed to show any significant improvement in fatigue or function compared to the placebo.	Constipation and diarrhoea (1 of each)
[100]	Randomised phase III clinical trial (5 arm)	Patients with cancer cachexia *n* = 332; LC arm *n* = 88	Advanced stage; loss of >5% of ideal or pre-illness weight	Randomised to one of 5 arms LC arm received 4 g/day	16 weeks	LC did not increase LBM, decrease REE or fatigue GPS improved 1.2 ± 0.76 ^a^ to 0.9 ± 0.86 ^a^ (*p* = 0.03); ECOG PS improved 1.88 ± 0.88 ^a^ to 1.5 ± 0.9 ^a^ (*p* = 0.0001)	Grade 3 or 4 diarrhoea (2)
[99]	Prospective multi-centre, placebo-controlled, randomised and double-blinded trial	72 pancreatic cancer patients; Evaluable *n* = 26 LC *n* = 14 PC *n* = 12	Advanced pancreatic cancer 10% weight loss in previous 6 months KPS > 60	Oral LC 4 g/day or placebo	12 weeks	BMI increase LC: 3.4% ± 1.35 ^c^ vs. PC: −1.5% ± 1.4 ^c^ *p* < 0.018; After 12 weeks difference was 4.9% ± 1.9 between groups *p* < 0.05; BIA showed improved body composition (*p* < 0.013) and BF (*p* < 0.041); EORTC-QLQ-C30/PAN26; Improved cognitive function LC: 0.30 versus PC-0.13 (*p* < 0.034), global health status LC 0.76 versus PC-0.32 (*p* < 0.041), and gastrointestinal symptoms improved-0.35 while participants on the placebo deteriorated 0.78 (*p* < 0.033).	
[101]	Phase III, randomised, double-blind, placebo-controlled trial followed by an open-label trial	*N* = 376 LC *n* = 189 PC *n* = 187	Invasive malignant cancer and moderate to severe fatigue	Week 0–4: 2 g oral L-carnitine or placebo; week 5–8: 2 g oral L-carnitine	4 weeks randomised, double-blind placebo, 4 weeks open label	33% of participants had LC deficiency; total mean carnitine increased 46.3 (95% CI, 44.1 to 48.4 µM/L) to 66.2 µM/L (95% CI, 62.4 to 69.9 µM/L); PC mean total carnitine increased nominally from 43.6 (95% CI, 41.4 to 45.7 µM/L) to 43.7 (95% CI, 40.7 to 46.7 µM/L; *p* = 0.004); No measures showed significant improvement when treatment and placebo arm compared, but significant improvement versus baseline suggesting a placebo effect.	Haemoglobin (LC-10, PC-9); Platelets (PC-1); Fatigue (LC-2); Itching /rash (LC-2); Death /disease progression (PC-2); Constipation (LC-1); Diarrhoea (LC-1, PC-4); Nausea /vomiting (LC-5, PC-6); Urinary tract infection (LC-1); Abdominal pain (LC-1, PC-2); Atrial fibrulation (PC-1); Dyspnea (2); Patient odour (PC-1); Extrapyramidal movement (PC-1); Headache (PC-1)

Abbreviations: BIA—Bioelectrical impedance analysis; BF—Body fat; BFI—Brief Fatigue Inventory; BMI—Body mass index; CES-D—Center for Epidemiological Studies Depression Scale; CI—confidence interval; CRP—C-reactive protein; ECOG PS—Eastern Cooperative Oncology Group performance status scale; EORTC-QLQ-C30/PAN26 European Organization for Research and Treatment of Cancer Quality of Life Questionnaire C30 with a pancreatic cancer-specific module PAN26; EQ-5DVAS—Global health status by the EQ-5D visual analog scale; ESS—Epworth Sleeplessness Scale; GPS—Glasgow Prognostic Score; LBM—Lean body mass; KPS—Karnofsky Performance Status; LC—L-Carnitine: MFSI-SF—Multidimensional Fatigue Symptom Inventory—Short Form; QoL—Quality of life; PC—Placebo-controlled; REE–Resting energy expenditure; ROS—Reactive oxygen species; ^a^—Mean ± SD, ^b^—Median (min, max), ^c^—Mean ± SEM.

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
