# Peer review of "Pancreatic Cancer Cachexia: The Role of Nutritional Interventions"

_healthcare, 2019, doi:10.3390/healthcare7030089_

Round 1

Reviewer 1 Report

This is a comprehensive paper written on an important topic. The tables are very helpful.

Here is my concerns:

1- This is a narrative not a systemic review. This should be stated clearly. 

2- There is not the best fit / match with the journal. The paper is not on the outcomes or health system but the biological mechanisms. I would suggest the paper should be sent somewhere else.

3- There are many number of paragraphs and text within the paragraphs that need citation. 

4- There are many short paragraphs.

5- There is considerable amount of redundancy in the text. To give one example, the text of the paragraphs 1 and 2 of the introduction have some repeated points. 

Author Response

Reviewer 1

Thank you for constructive comments. We have addressed each of your comments below.

1-     This is a narrative not a systemic review. This should be stated clearly. 

We have edited the manuscript to ensure the reader will understand that the manuscript is a narrative and not a systemic review.

Abstract Line 21 – the word “review” was removed.

Introduction Lines  63, 185-186 “In this narrative, …..”

2-     There is not the best fit / match with the journal. The paper is not on the outcomes or health system but the biological mechanisms. I would suggest the paper should be sent somewhere else.

The focus of the manuscript has been modified at the suggestion of Reviewer 2. There is now less of a focus on mechanisms and more of a focus on nutrition and outcomes.

3-     There are many number of paragraphs and text within the paragraphs that need citation. 

Additional statements have been cited. We are willing to include other citations if you could please point out where this is necessary.

4-     There are many short paragraphs.

We have combined many of the short paragraphs e.g. paragraphs 1 and 3. We have removed section 2, where many of the short paragraphs could be found, and incorporated some of it into the introduction.

5-     There is considerable amount of redundancy in the text. To give one example, the text of the paragraphs 1 and 2 of the introduction have some repeated points. 

The text has been carefully checked for redundancy, and the necessary text removed.

Reviewer 2 Report

There is a need for a publication such as this summarising the status of nutritional interventions, although it could be more broadly based rather than restricted to pancreatic cancer

This paper is about nutritional interventions to support cancer cachexia management and not as the title suggests about Mechanisms and the role of nutrition.  Consistent with this, the section on mechanisms is superficial and deals with a highly selected number of molecules.  For example, it doesn’t mention GDF15 or Tweak, for which there is far better evidence than that sighted by the authors.  Also no mentioned is the importance of psychological factors and the role of chemotherapy and radiotherapy.  Further, in this section, I don’t believe that mechanisms can be restricted to pancreatic cancer cachexia, need to deal with cancer cachexia more generally and be focused heavily on in-vivo data.  As this is essentially an introductory section, one solution might be to shorten it, make it more general, and include it within the introductory paragraph. 

I believe that the title of this manuscript should be altered to reflect its real emphasis, which is nutritional intervention to support pancreatic cancer cachexia management.  The title could go something like Pancreatic Cancer Cachexia: Nutritional Interventions. 

Page 5, paragraph 2.  It is not completely accurate to say that the cause of calorie intake reduction in patients with cancer cachexia is not well understood. Chemotherapy, radiotherapy and disruption of the gastrointestinal tract are all potent causes. Additionally, there is very good evidence for the role of at least one anorexic cytokine, GDF15, commonly over expressed in cancers in general and pancreatic cancer specifically. 

Page 5, section 3.2. The biggest impediment to oral nutritional supplementation is likely to be anorexia.  It is very hard to get profoundly anorectic people to eat, and this point should be made.

Whilst I am not a proponent of this view, the current, most widely accepted definition of cancer cachexia is that it must be resistant to dietary intervention, so this needs to be acknowledged.  

General.I would recommend a sentence or two at the end of each of the subsections summarizing the overall results / views /  status.  This has been done in some sections e.g. 3.6 but would be useful if done in more sections. 

What about a section on parentral supplementation

Author Response

Thank you for your constructive criticisms. We have followed your suggestions and edited the manuscript (and title) to focus on nutrition rather than mechanisms. Section 2 was deleted, and where relevant, has been included in the Introduction. The changes in the first half of the manuscript are extensive and can be viewed as tracked changes.

Page 5, paragraph 2.  It is not completely accurate to say that the cause of calorie intake reduction in patients with cancer cachexia is not well understood. Chemotherapy, radiotherapy and disruption of the gastrointestinal tract are all potent causes. Additionally, there is very good evidence for the role of at least one anorexic cytokine, GDF15, commonly over expressed in cancers in general and pancreatic cancer specifically. 

All authors agree that the reduction in calorie intake is largely understood, and the text has been edited accordingly (Lines 207-209 have been removed). GDF-15 has been included in the context of nutrition and cachexia (Lines 97-100; 283-288; 442-444). Thank you for this suggestion.

Page 5, section 3.2. The biggest impediment to oral nutritional supplementation is likely to be anorexia.  It is very hard to get profoundly anorectic people to eat, and this point should be made.

Whilst I am not a proponent of this view, the current, most widely accepted definition of cancer cachexia is that it must be resistant to dietary intervention, so this needs to be acknowledged.  

We have acknowledged that many authors regard cachexia as resistant to nutrition (Lines 62-64). We have mentioned anorexia in the section on parenteral nutrition (Lines 260-261).

General.I would recommend a sentence or two at the end of each of the subsections summarizing the overall results / views /  status.  This has been done in some sections e.g. 3.6 but would be useful if done in more sections. 

We agree with your suggestion. However, Review 1 has commented on the need to remove repetition, which was not entirely evident to me. For this reason we have not included summary statements. Should the Editor agree that the inclusion of such statements would enhance the manuscript, we will do so.

What about a section on parentral supplementation?

Thank you for the suggestion. This was an omission on our part. A paragraph on parenteral nutrition has been included in section 2.2 (Lines 260-268)

Round 2

Reviewer 1 Report

The authors have addressed all of my comments. The paper now only needs to work on its list of limitations. Not being a systematic review, not having critical appraisal of the studies that were reviewed, etc, all need to be discussed in a paragraph with the tittle limitation. 

Author Response

Thank you for your comment. We have included a section on limitations, as requested. This section can be found immediately prior to the Conclusion. Other minor changes have been made and all changes have been tracked.

Reviewer 2 Report

Th authors have made most of the suggested changes

Author Response

A section on limitations has been included and numerous minor edits have been made. All changes have been tracked.